# Federated learning with LSTM for intrusion detection in IoT-based wireless sensor networks: a multi-dataset analysis

Raja Waseem Anwar[1], Mohammad Abrar[2], Abdu Salam[3] and Faizan Ullah[4]

[1] Department of Computer Science, German University of Technology in Oman, Muscat, Oman
[2] Faculty of Computer Studies, Arab Open University, Muscat, Oman
[3] Department of Computer Science, Abdul Wali Khan University, Mardan, KPK, Pakistan
[4] Department of Computer Science, Bacha Khan University, Charsadda, KPK, Pakistan



Corresponding author
Raja Waseem Anwar,
raja.anwar@gutech.edu.om

## ABSTRACT

Intrusion detection in Internet of Things (IoT)-based wireless sensor networks (WSNs) is essential due to their widespread use and inherent vulnerability to security breaches. Traditional centralized intrusion detection systems (IDS) face significant challenges in data privacy, computational efficiency, and scalability, particularly in resource-constrained IoT environments. This study aims to create and assess a federated learning (FL) framework that integrates with long short-term memory (LSTM) networks for efficient intrusion detection in IoT-based WSNs. We design the framework to enhance detection accuracy, minimize false positive rates (FPR), and ensure data privacy, while maintaining system scalability. Using an FL approach, multiple IoT nodes collaboratively train a global LSTM model without exchanging raw data, thereby addressing privacy concerns and improving detection capabilities. The proposed model was tested on three widely used datasets: WSN-DS, CIC-IDS-2017, and UNSW-NB15. The evaluation metrics for its performance included accuracy, F1 score, FPR, and root mean square error (RMSE). We evaluated the performance of the FL-based LSTM model against traditional centralized models, finding significant improvements in intrusion detection. The FL-based LSTM model achieved higher accuracy and a lower FPR across all datasets than centralized models. It effectively managed sequential data in WSNs, ensuring data privacy while maintaining competitive performance, particularly in complex attack scenarios. FL and LSTM networks work well together to make a strong way to find intrusions in IoT-based WSNs, which improves both privacy and detection. This study underscores the potential of FL-based systems to address key challenges in IoT security, including data privacy, scalability, and performance, making the proposed framework suitable for real-world IoT applications.

## INTRODUCTION

Within the Internet of Things (IoT) framework, wireless sensor networks (WSNs) offer great potential for several areas, including healthcare, agriculture, and the military.

Nevertheless, implementing WSNs in open and/or resource-scarce environments raises several security issues. Such networks include small, low power sensors that use wireless communication links for networking and hence are very susceptible to security attacks when interfaced with the IoT systems. As a result, a sound and secure mechanism in WSNs, especially for IoT-based applications, is considered a challenging concern. To mitigate these threats, intrusion detection systems (IDSs) have been integrated to run constantly in the network to detect intrusions and produce an alert when intrusions are suspected. This active monitoring enhances the general security of WSNs and reduces the probability of risks arising from the application of WSNs in sensitive areas (*Almomani, Al-Kasasbeh & Al-Akhras, 2016*). WSNs are systems of sensors dispersed geographically that collect data about their environment and relay the details wirelessly and are an important component in IoT. They allow real-time data collection and monitoring in various sectors, making IoT applications more effective. In IoT-based WSNs, an entire network of sensor nodes is deployed to collect data and transmit it to a base station (BS) or other IoT devices to make decisions. Integrating IoT with WSNs enhances the opportunities of creating 'smart' cities, distant observation and monitoring, and environmental monitoring, considerably increasing the effectiveness and performance (*Akyildiz, 2002*). The security issue in WSNs is embedded since they possess technical features: low power supply, processing abilities, and memory. Such constraints hinder the traditional security measures such as Firewall and encryption techniques to be applicable on WSN. As a result, effective security solutions require developing specific security solutions aligned with WSN demands (*Ghazal et al., 2023*; *Anwar et al., 2019*). The significant adoption of IoT devices and sensors amplifies the threat exposure on systems due to multiple threats like DoS attacks, spying, and data amendment (*Abuserrieh & Alalfi, 2024*). WSNs are typically implemented in sensitive applications such as health and defense; hence, their security should be protected. However, using WSNs in open and possibly adversarial environments introduces other layers of protection needed for the WSNs. Sensor nodes are vulnerable to various attacks, affecting the different zones above and causing false data injection or impediment of normal message transmission. Thus, it is highly important to ensure the existence of robust yet remaining light and sophisticated security measures for IoT-based WSNs, considering the data intensity and privacy measures (*Yalli, Hasan & Badawi, 2024*). IDS are required for WSNs as these networks require additional layers of protection apart from basic prevention mechanisms that prevent an attacker from breaking into the WSN. We also classify Intrusion detection systems based on the IDS scenarios as Signature-based and Anomaly-based. The signature-based IDS compares network activity against expected behavioral patterns to identify unconventional patterns that can only be attributed to an intrusion. Using this approach, one can identify familiar attack patterns, but there is also the likelihood of receiving numerous alerts for those patterns that are deemed to exist by the system, even in the absence of an act of intrusion (*Ullah et al., 2020*). A signature-based IDS is an ID system that works through the detection of declared attack signatures or trends. Although this method allows recognition of well-known threats, it cannot recognize brand-new types of attacks and thus is not very flexible to the newly created types of threats (*Hamza et al., 2024*).

This consequently poses a major challenge to IDS in WSN due to the constraint of available resources in sensor nodes. However, introducing integrated and real-time IDS in WSNs remains a tough and incompatible factor with the WSNs architecture. Consequently, IDS for WSNs must be lightweight and perform efficient operation while respecting the fact that sensor node platforms have limited processing ability and energy supply (*Roman, Najera & Lopez, 2011*). Current research has shifted toward applying ML and DL models, particularly long short-term memory (LSTM), in optimizing the IDS setup in WSNs. These algorithms are optimally designed to integrate with today's ever-changing networks and improve detection ability while reducing as much as possible false alarms (*Roman, Najera & Lopez, 2011*). Federated learning (FL) can become an advantage if integrated into WSN–IDS when it is aimed at strengthening WSNs using IoT parts. FL supports distributed model training, so during the training process, each node can participate in developing a single model while not uploading the raw data, which can reduce the risk of a particular key point for an attack against the central system. As in the case of IoT-based WSNs, centralized IDS cannot meet the required specifications of a network. One of which is the fact that WSN nodes suffer from minimal computation, memory and power resources. The existing centralized IDS designs receive massive amounts of data at one central server, and this causes traffic jams and exhausts the sensors of the nodes (*Hajiheidari et al., 2019*). This characteristic makes gross and comprehensive solutions unworkable and counterproductive in expansive IoT networks as they produce unrelenting data from sundry smart appliances. When the number of nodes within an IoT infrastructure increases, a centralized IDS can face increasing amounts of data to analyze and the need to cover a larger geographical territory to identify threat activity. This high device volume also puts a lot of pressure on the central server to process data and analyze them; hence, real-time threat detection may prove to be a big challenge due to the increased data volume that the system can hardly handle (*Roman, Alcaraz & Lopez, 2007*). Latency is a significant problem—instead of sending all data to a central point for processing, it takes time. This can be very important in health or industrial applications where the response to a command cannot wait for the next few seconds.

As evident, FL provides a more feasible approach to address some of the challenges of centralized IDS in IoT-based WSNs. This is in contrast to a model where only the server is involved in processing since FL allows each node or device within a network to carry out model training on the data owned by each of them. This process forms local aggregate models without transferring raw data to form a global model. Unfortunately, this approach reduces the amount of data transferred over the network and increases privacy by about half (*Kairouz et al., 2021*). FL reduces the communication between nodes and the central server and, therefore, helps to save energy and bandwidth, which is suitable in WSNs. Every node performs some computations independently and only sends updates in the network. This relieves network traffic and increases the sensor node lifetime, scalability, and efficiency of the system on the total (*Bhatti et al., 2024*). FL also reduces the probability of instances of susceptibilities to specific attack points. Since model training happens simultaneously at some nodes, the impact of a node controlled by a malicious actor is not tremendously significant. Also, the distributed model of FL makes it difficult for an

attacker to tamper with the functionalities of this system, thus improving the reliability and robustness of the IDS (*Tonin et al., 2024*). In addition, FL, because of its distributed structure, can smoothly incorporate new devices and nodes into IoT-based WSNs. In this sense, each new node engages in the learning process, contributing to less demand for central server performance. It also improves the flexibility of the security solution, which makes it possible to extend the solution as the size of the network increases (*Salam et al., 2024*).

LSTMs are highly suitable for intrusion detection in IoT-based WSNs due to their ability to process sequential patterns in network traffic data effectively. Their architecture is specifically designed to model long-term dependencies, enabling them to capture temporal patterns in intrusion behaviors, such as Distributed Denial of Service (DDoS) attacks and other multi-step intrusion scenarios. This capability allows LSTMs to distinguish between normal and abnormal network activities with high accuracy, making them particularly effective for detecting complex, evolving attack patterns in dynamic IoT environments.

The article presents an IDS for IoT-based WSNs that integrates FL with LSTM networks as an effective security solution. The main contributions of this work include:

- It addresses the limitations of centralized IDS by utilizing decentralized FL, which enhances scalability and reduces the risk of single points of failure.
- Integrating LSTM enhances detection accuracy by learning from sequential data, making it highly effective against sophisticated attacks in WSNs.
- This study adopts a multi-dataset approach, utilizing WSN-DS, CIC-IDS-2017, and UNSW-NB15 to comprehensively evaluate and ensure robust detection performance across diverse IoT network scenarios.
- Several metrics, including accuracy, F1 score, FPR, root mean square error (RMSE), and a confusion matrix, were used to evaluate the proposed approach's performance. This comprehensive set of metrics provides detailed insights into the system's effectiveness and reliability.

These contributions enhance the security of IoT-based WSNs' operating environment by improving the reliability and effectiveness of intrusion detection systems. The rest of the article is organized as follows: the "Related Work" section reviews related work. "Datasets and Preprocessing" describes datasets and preprocessing. "Proposed Methodology" explains the proposed FL-based LSTM framework and outlines the experimental setup. "Results and Analysis" presents the results and analysis. "Discussion" concludes the article and suggests future research directions.

## RELATED WORK

As insecurity incidents increase in IoT and WSNs, the search for an IDS for these networks grows exponentially. However, because of key constraints such as extensive resource constraints and the distributed nature of the IoT and WSN architectures, typical IDS approaches present recognizable problems in these networks. To solve these problems,

researchers have attempted to study various methods, including traditional digital signature techniques and new artificial neural networks, to make intrusion detection more accurate and relevant to functioning networks with limited available resources.

## IDS in IoT and WSN

IDS are useful in other unrecognized activities unallowable in IoT-based WSNs. An IDS resides centrally on a network and monitors the traffic to ensure it is not being infiltrated or processed erratically. In WSNs and IoT paradigms, IDS must operate under the premise of adjusting to the capabilities of sensor nodes, including CPU, memory, and energy. There are two basic categories of IDS: The signature-based and the anomaly-based IDS, respectively, and there are some differences of advantages and disadvantages between the two.

Machine and deep learning have recently been used to enhance the detection of accurate results while reducing computational weight (*Jyothsna, Prasad & Prasad, 2011*). Signature-based IDS works on the principle of identifying an intrusion by matching its signatures to that of a list of recorded ones (*Kumar & Kumar, 2023*). This method is very efficient at finding previously encountered threats, often accompanied by a significantly few false positives, because this method utilizes a set of defined signatures to look for specific types of attacks. Most existing systems are signature-based because of the operations' simplicity and high accuracy regarding previously known threats. Their main drawback is their failure to identify new or developing attacks since their efficacy depends on their signatures database. The frequent update of the signature database due to the development of new threats can prove to be problematic in terms of time and again pinning down the fact that IoT and WSN have limited bandwidth and availability of storage space (*Ananthakumar, Ganediwal & Kunte, 2015*). However, in anomaly-based IDS, intrusions are identified based on deviations from the expected behavior of the network (*Garcia-Teodoro et al., 2009*; *Anwar & Abdullah, 2023*). This approach closely approximates the expected network traffic and alerts on traffic that deviates from predicted norms. Anomaly-based IDS helps identify currently unseen or brand-new types of attacks because an IDS of this kind has no predefined pattern to search for. This makes them more suited to changes in threats in IoT-based WSNs. Nevertheless, this results in higher FPRs as normal network traffic fluctuations may occasionally be considered attacks (*Sedjelmaci, Senouci & Al-Bahri, 2016*). Furthermore, anomaly-based systems can be expensive in terms of computational complexity and trivial for WSNs since resources are scarce. Nevertheless, they are helpful for modern IDS solutions because they can respond to new threats.

Modern advancements in machine learning (ML) and deep learning (DL) have created a new generation of IDS for IoT and WSN. Such systems employ statistical data models to understand the network traffic better and detect intrusions more effectively. More complex heuristic-based supervised learning algorithms, such as SVM and RF, have also been implemented to categorize network activities as normal or malicious (*Tsimenidis, Lagkas & Rantos, 2022*). Further, DL models like CNN and RNN have given commendable results in managing intricate sequential data in WSNs where attack behaviors may transform with time (*Gebremariam, Panda & Indu, 2023*). The newer variety of RNNs known as LSTM

networks has enhanced intrusion detection due to their ability to learn from past experiences, thus identifying long-term dependencies in traffic patterns. LSTM-based IDS can identify complex attacks like DDoS and wormhole attacks, as these are longer processes (*Salmi & Oughdir, 2023*). Machine learning and deep learning models also have an advantage in constantly updating after learning more threats than other models, making them more robust in dynamic IoT conditions. Albeit efficient, ML and DL approaches involve high computational complexities, which still present a considerable limitation, especially in constrained WSNs. However, due to these considerations, the authors report that lightweight models have been combined with FL techniques that enable distributed model training. Thus, ML and DL solutions are becoming more and more feasible for practical WSN security (*Bonawitz, 1902*).

## Federated learning for IoT security

Federated learning (FL) has been disclosed as an innovative method for improving security in IoT scenarios. Unlike the paramount centralized models of machine learning, which presuppose data to be transferred to a central server, FL presupposes models' training at several devices or nodes simultaneously without exchanging raw data. This improves the decentralized nature of the IoT-based WSNs since traditional central security is ineffective due to resource and privacy limitations. It is also particularly challenging in an IoT setting where devices produce large volumes of data to transfer all this data to a central server for processing, as this will burden the network bandwidth, result in high latency, and consume a device's energy (*Tran et al., 2019*). FL addresses these issues by empowering nodes to perform and train their models on their own data buckets. Client-side: Only the model updates, such as weights, are uploaded to the central server, where they are summed to constitute a global model. This approach helps to decrease the flow of data across the network as well as to preserve the confidentiality of the information produced by IoT devices since raw data are retained on the local devices (*Kairouz et al., 2021*).

FL also addresses a critical challenge in IoT security: the heterogeneity of devices and environments is a decisive factor in determining networks' performance and the quality of provided services. In a conventional IoT network, there are connected devices, such as smart handheld devices and power-limited sensors. Every device is located in a different context and deals with various threats. FL enables each device to develop a model using data and security settings of the local device, making the models developed more diverse and adaptive to the settings of IoT devices (*Li et al., 2020*). However, FL strengthens IoT security by eliminating the bootstrapping problem, which poses multiple attacks and single points of failure, which are always a severe weakness in centralized systems. Since the model training is divided among several devices, the intrusion into one or several devices represents no threat to the whole network (*Bonawitz, 1902*).

In the case of IoT security, another key positive attribute of FL is that it is well-suited to real-time work. Highspeed identification of threats and their containment is important for many IoT uses like smart healthcare or industrial automation. FL enables IoT devices to

identify and fight attacks at the edge without adding the latency of interacting with a central server. This decentralization helps in the fast determination of threats and saves time for corrective action, which is suitable for applications where time is of the essence (*Neto et al., 2023*).

As much as FL has long offered a relatively promising way to enhance the security of IoT, there are several limitations to its use as well. There are still issues in active research, including the communication overhead associated with the continual need to update models, the need to coordinate between devices, and the susceptibility of the global model to adversarial influence. Challenges like privacy leakage, malicious aggregation, and network disruption are some of the major concerns that have cropped up in FL, and some solutions under implementation include secure aggregation, differential privacy, and Byzantine resilient algorithms for making FL more robust and safe for IoT purpose (*Chen et al., 2023*).

## LSTM for sequential data in network security

Recurrent neural networks (RNNs) with LSTM are a very popular approach in network security because they deal with sequential data. Unlike feed-forward networks, LSTMs are developed to work with long-term dependencies in data, which is essential in applications where the order of events is important: analysis of network traffic and intrusion detection (*Kim et al., 2016*). This ability to process data sequences makes LSTMs good at learning about temporal patterns, which are crucial when detecting new and emerging attack techniques in WSNs and IoT.

The network traffic is twofold sequential because packets, transactions, and other elements forming the network traffic are ordered. Most attack patterns, such as those engaged in a DDoS attack, the wormhole attack, or the replay attack, are multistep. Due to the use of memory cells and gates, LSTMs are efficient in these cases as they can learn and recognize these long sequences of activity deviations that other models cannot detect, for they lack temporal awareness (*Yin et al., 2023*; *Anwar & Qureshi, 2023*). Since LSTM models keep information over past times, they can better distinguish between normal and evolving intrusions than the other models that work on static snapshots of network traffic data.

By design, LSTM networks possess one of the biggest strengths: the ability to effectively model short- and long-term dependencies. In network security, it is possible to identify quick and frequent attacks and slow and long-lasting ones. For instance, in DDoS attacks, traffic tends to be flooded quickly, which LSTM can classify as an anomaly because of the sequence of un/packet flows (*Ullah et al., 2022*). While others may take months or even weeks to penetrate a network, especially given that APTs are known to gather as much information as possible.

LSTM can learn and detect such anomalies by observing anomalous patterns that span long durations. In recent studies, LSTMs have been incorporated into IDS in WSNs and efficiently detect a wide range of complicated attack paths that are invisible to traditional IDS. It is based on the finding that in network traffic data LSTM models, detection rates of

**Table 1 Comparative analysis of LSTM, GRU, and CNN for intrusion detection in IoT-based WSNs.**

| Model | Strengths | Weaknesses | Performance in intrusion detection |
|---|---|---|---|
| LSTM (*Hochreiter, 1997*) | Excellent at capturing long-term dependencies in sequential data; well-suited for modeling temporal patterns. | Computationally intensive; requires more training time and resources. | Achieves the highest accuracy and precision in detecting multi-step attacks (*e.g.*, DDoS) and temporal anomalies. |
| GRU (*Chung et al., 2014*) | Computationally efficient; faster training and lower memory requirements compared to LSTM. | Less effective in modeling long-term dependencies in complex temporal sequences. | Performs reasonably well but struggles with detecting prolonged or complex sequential intrusion behaviors |
| CNN (*LeCun, Bengio & Hinton, 2015*) | Highly effective for extracting spatial features; computationally efficient for non-sequential data. | Not designed for sequential modeling; lacks temporal awareness crucial for detecting evolving attack patterns. | Performs well on static features but fails to identify dynamic, time-dependent intrusion patterns in network traffic data. |

diverse attacks are high, such as Probe, remote-to-local (R2L), and user-to-root (U2R) attacks (*Liao et al., 2024*). The possibility of processing historical data truly helps in intrusion detection, as LSTM models can rely on a network's history of activity and learn about malicious activities in the future.

Nonetheless, using LSTM in environments such as WSNs has been challenging since LSTM networks have high computational and memory costs. To this, literature research has looked at optimizing LSTM with lightweight implementations that could run in WSN without straining system resources. Such optimization measures are pruning technique and quantization, which help to bring down the size of LSTM, affecting the detection accuracy least (*de Souza et al., 2024*). In addition, combining LSTM with FL can even enhance LSTM's applicability in WSN more because FL allows model training on each node to decrease the amount of data transferred and increase privacy (*Hulayyil, Li & Xu, 2023*).

Table 1 highlights the strengths, weaknesses, and performance of LSTM (*Hochreiter, 1997*), GRU (*Chung et al., 2014*), and CNN (*LeCun, Bengio & Hinton, 2015*) models in intrusion detection, emphasizing LSTM's suitability for sequential and temporal data.

# DATASETS AND PREPROCESSING

This section discusses the three key datasets used in our research: It was performed in WSN-DS, CIC-IDS-2017, and UNSW-NB15 datasets. These datasets are important in assessing the performance of the proposed Intrusion Detection System (IDS), especially in determining different types of attacks in IoT-based WSNs. In addition to these datasets described in the following section, we also discuss the basic preprocessing required to prepare the data for training and testing.

## Datasets

The datasets used in this article hold diverse network traffic and attack characteristics. We can construct a more accurate model by having two types of datasets since each type is differently structured. Table 2 gives each dataset a brief description.

**Table 2 Summary of datasets with records, features, and attack types.**

| Dataset | Data type | Number of records | Number of features | Attacks included |
|---|---|---|---|---|
| WSN-DS | Simulated WSN | 374,661 | 23 | Blackhole, Grayhole, Flooding, Scheduling |
| CIC-IDS-2017 | Real Network Data | 2,830,743 | 80 | DoS, DDoS, Brute Force, Infiltration |
| UNSW-NB15 | Hybrid Network Data | 2,540,044 | 49 | Fuzzers, Analysis, Backdoors, Worms |

### WSN-DS dataset

For intrusion detection in WSN, the author introduced a WSN-DS dataset. It contains data from simulation experiments in the Network Simulator 2 (NS2) environment. The dataset includes four major types of Denial of Service (DoS) attacks: Blackhole, Grayhole, Flooding, and Scheduling attacks. WSN-DS will analyze the LEACH routing protocol as this has been the preferred routing technique for WSNs because of the low energy consumption. The dataset has 23 attributes in this experiment, including network parameters like received signal strength indication (RSSI) energy consumed, distance to cluster head, *etc*. These features signify normal and attack behavior in the network (*WSN-DS, 2016*).

For WSN-DS, we excluded records that seemed unrelated or contained missing information during data preprocessing. All the values were scaled down or up because features were normalized to make the scales of all data points as similar as possible. The set was divided into training and testing data to equalize the attack distribution and normal instances.

### CIC-IDS-2017 dataset

The CIC-IDS-2017 dataset is one of the largest intrusion detection datasets, covering five days of real network traffic. It mocks modern attacks like DDoS, Brute Force, Web, and Infiltration. The targeted observed network features amount to 80 and cover a wide set of characteristics: flow duration, packet size, time between packets, *etc*. The dataset is organized so that each record is labeled as either normal traffic or belonging to a particular type of attack (*CIF Cybersecurity, 2017*). Since the CIC-IDS-2017 dataset is large, preprocessing included sampling to extrude most of it to make training time reasonable. We also used feature selection to make the dataset less dimensional to identify the most important attributes for intrusion detection. A label encoding strategy was employed to transform categorical data, especially the attack types, into numerical data for modeling.

### UNSW-NB15 dataset

The UNSW-NB15 dataset is real-world network traffic with stimulation of attack conditions. Nine distinct attack kinds are included: Fuzzers, Backdoors, Worms, and more, which is a primary representation of possible network weaknesses. The preprocessed data contains 49 features that characterize different aspects of network traffic and is used to differentiate between network traffic generated by an abusive node and normal behavior

(*UNSW, 2015*). In the case of UNSW-NB15 log files, we have treated empty values and further normalized the feature set. However, this article also employed some techniques of handling class imbalance to enhance the classifiers' effectiveness so that the models understood the attack types when training them.

## Data preprocessing

Data preprocessing is a critical step in transforming raw data into a form suitable for modeling, ensuring that the datasets are aligned with the unique challenges of IoT-based WSNs. The three datasets used in this study—WSN-DS, CIC-IDS-2017, and UNSW-NB15—underwent comprehensive preprocessing steps, including feature extraction, normalization, and class balancing. These steps were designed to address IoT-specific challenges such as resource constraints, heterogeneous data distributions, and the need for efficient analysis of low-bandwidth, high-variability traffic patterns. For the WSN-DS dataset, preprocessing involved removing irrelevant or incomplete records and normalizing network parameters, such as RSSI and energy consumption, to reflect the constrained nature of WSN environments. CIC-IDS-2017, a more diverse dataset, required dimensionality reduction and categorical label encoding to handle its high-dimensional structure and ensure compatibility with sequential modeling. UNSW-NB15, known for its sophisticated attack scenarios, underwent additional steps to balance attack and normal instances, mitigating the class imbalance common in IoT-related datasets.

### Data cleaning and handling missing values

Data cleaning is an initial phase of data preprocessing that focuses on removing duplicities, inconsistencies, or irrelevant data. One part of this process involves treating missing values, which can be typical for several reasons, including but not limited to malfunctions of the ships' sensors or corruption of the data. Sparse data may be a moot point because multiple machine learning algorithms require assistance in processing datasets with complete data.

We used the mean, median, and mode imputation methods to address missing values. For all numerical features, missing values were substituted by the mean or median of this feature to maintain the data's distribution.

$$X_{new} = \frac{1}{n} \sum_{i=1}^{n} X_i \tag{1}$$

where $X_i$ is the value of the feature and $n$ is the number of available (non-missing) values.

For categorical features, missing values were filled with the mode, *i.e.*, the most frequent value in the feature column. In some cases, rows with excessive missing values were dropped to prevent introducing too much bias into the dataset.

### Feature extraction and selection

Feature extraction involves transforming raw data into informative features, while feature selection helps identify the most relevant features for model training. Given the high

dimensionality of our datasets, selecting the right features was crucial for reducing model complexity and improving performance.

The following methods were used for feature selection:

- Correlation matrix: We computed the correlation coefficient $\rho$\rho$\rho$ between pairs of features to identify and remove highly correlated features, where:

$$\rho(X, Y) = \frac{\sum (X_i - \bar{X})(Y_i - \bar{Y})}{\sqrt{\sum (X_i - \bar{X})^2} \sqrt{\sum (Y_i - \bar{Y})^2}}. \tag{2}$$

Features with a correlation coefficient $|p| > 0.9$ were considered redundant.

- Recursive feature elimination (RFE): RFE was applied to rank features by importance. The algorithm recursively eliminates the least important features based on model performance.
- Principal component analysis (PCA): PCA was used to reduce dimensionality while preserving variance. PCA transforms the dataset into a set of linearly uncorrelated components:

$$Z = XW \tag{3}$$

where $X$ is the original data, $W$ is the matrix of principal components, and $Z$ is the transformed data with reduced dimensions.

### Normalization and standardization

To ensure that features with different scales do not disproportionately influence the model, normalization and standardization were applied:

Normalization scales the data to a range between 0 and 1. This technique is beneficial for algorithms like neural networks that rely on distance-based calculations:

$$X_{norm} = \frac{X - X_{min}}{X_{max} - X_{min}} \tag{4}$$

where $X_{min}$ and $X_{max}$ are the minimum and maximum values of feature $X$.

Standardization rescales the data to have a mean of 0 and a standard deviation of 1, which helps with algorithms like support vector machines and logistic regression:

$$X_{std} = \frac{X - \mu}{\sigma} \tag{5}$$

where $\mu$ is the mean and $\sigma$ is the standard deviation of the feature.

### Dataset balancing techniques

Class imbalance is a common issue in intrusion detection datasets, where the number of normal instances often far exceeds the number of attack instances. Imbalanced datasets can lead to biased models that perform poorly in minority classes. To address this, we used the following balancing techniques: To create additional synthetic samples of

the minority class, the Synthetic Minority Over-sampling Technique (SMOTE) was applied. This method creates new instances by interpolating existing minority class samples:

$$X_{new} = X_{minority} + \lambda\big(X_{nearest\_neighbor} - X_{minority}\big) \tag{6}$$

where $\lambda$ is a random number between 0 and 1, and $X_{nearest\_neighbor}$ is the nearest neighbor of the minority sample.

We randomly reduced the number of majority class instances to balance the dataset and ensure the training process did not favor the dominant class. For algorithms that support it, such as decision trees and neural networks, we applied class weighting to assign higher weights to the minority class, reducing the bias towards the majority class during training.

## PROPOSED METHODOLOGY

FL is an algorithmic machine learning model involving several devices or nodes without directly sharing raw data. This approach is most suitable in IoT-based WSNs; privacy, bandwidth, and resource scarcities render using traditional centralized learning approaches problematic. In an FL framework, every node in the network builds a model on its own data set. Every node sends not the whole set of data, but only the updates to the model (for example, the weights) to a particular central server called an aggregator. The aggregator then integrates these updates to build a global model and disseminates it with all the nodes involved. This process is repeated over several federated rounds until the model at the global level meets optimal performance. Figure 1 presents a visual representation of the FL framework.

### LSTM architecture for intrusion detection

LSTM networks are RNNs developed to model long-duration dependencies in sequences. This makes LSTM appropriate for identifying intrusions in IoT-based WSNs where attacks occur in a temporal pattern. One serious drawback of traditional RNNs is the vanishing gradient problem, while LSTMs can accumulate information over long sequences, therefore identifying immediate and delayed attack behaviors. The key components of LSTM architecture is:

- **Memory cell:** LSTM's memory cell retains information occasionally. It can consolidate which information to recall or forget depending on the network's requirements, efficiently managing long data sequences.
- **Gates:** A basic LSTM employs three gates: the input gate, the forget gate, and the output gate. These gates regulate the data flow in and out of the memory cell. The forget gate controls which information from the previous memory state should be discarded.

$$f_t = \sigma\big(W_f \cdot [h_{t-1}, x_t] + b_f\big) \tag{7}$$

where $f_t$ is the forget gate's output, $h_{t-1}$ is the previous hidden state, $x_t$ is the input, and $W_f$ and $b_f$ are the weights and bias. Updates the memory cell with new information.

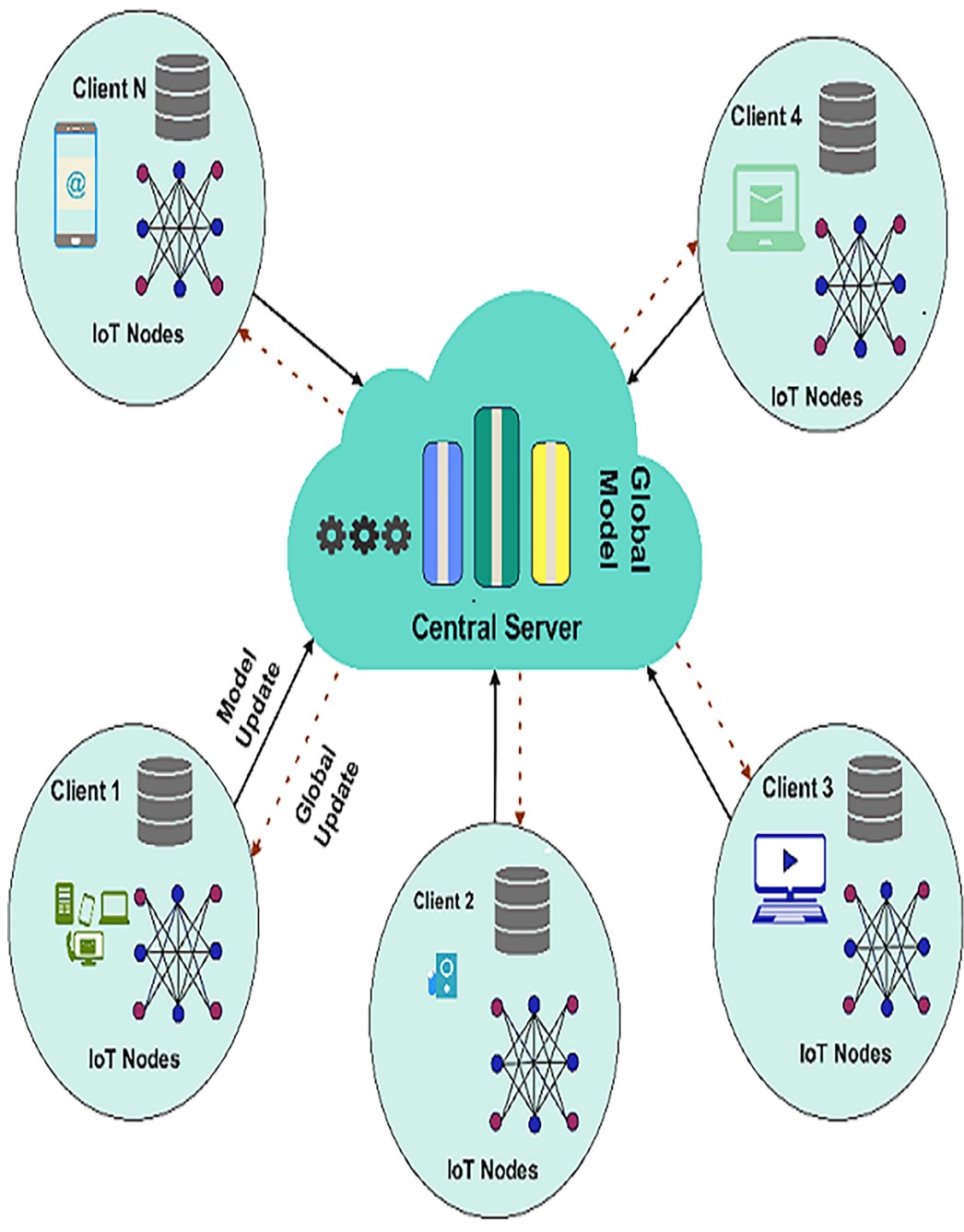

**Figure 1 Federated learning framework.**

$$i_t = \sigma(W_i \cdot [h_{t-1}, x_t] + b_i) \tag{8}$$

Determines the output based on the current memory and hidden states.

$$o_t = \sigma(W_o \cdot [h_{t-1}, x_t] + b_o) \tag{9}$$

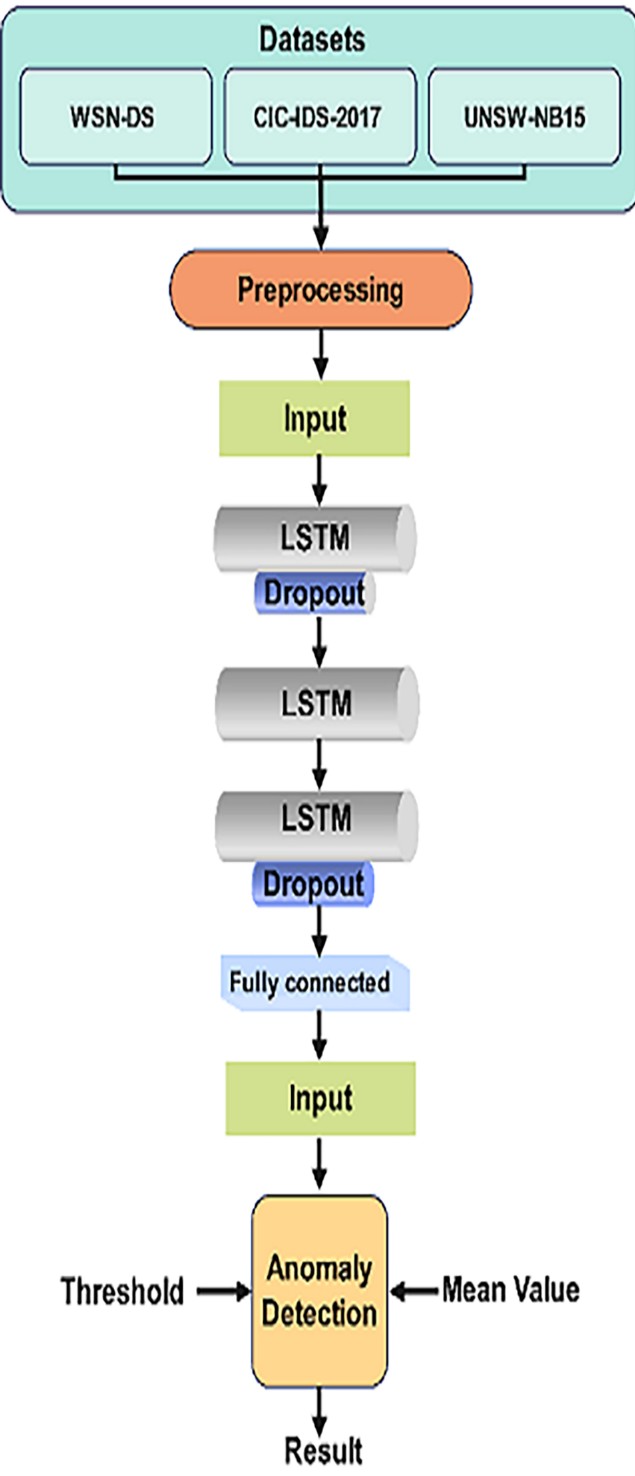

**Figure 2 LSTM architecture for intrusion detection.**

The cell state carries information through the network. It is updated at each step based on the information passed through the gates, allowing the network to retain crucial information from earlier steps.

LSTM networks are particularly effective in detecting anomalies or intrusions in WSNs by learning from network traffic sequences and identifying normal and abnormal behavior patterns over time. By analyzing the temporal dependencies, LSTM can recognize attacks like DDoS, probing, or slow-rate attacks, which unfold gradually. Figure 2 depicts the visual representation of LSTM Architecture.

This architecture efficiently detects network intrusions by learning from the sequence of network data, making it ideal for IoT-based WSNs where attacks evolve. LSTMs help to enhance detection accuracy, reduce false positives, and capture complex temporal patterns in network behavior.

## Integrating LSTM with FL

The integration of LSTM and FL enhances LSTM's ability to perform temporal information processing in addition to FL's decentralized, privacy-preserving training. This approach is especially helpful for intrusion detection in IoT-based WSNs since constellation and time series are essential for such networks. This architecture uses only three LSTM layers to balance the performance with computational efficiency. By using the limited number of layers coupling with dropout regularization, it reduces the overall memory usage, complexity, and computational cost. The lightweight architecture ensures that the model remains effective in the resource constraints IoT devices.

### *Local model training*

Thus, in an FL framework, every node, for instance, a sensor in a WSN, trains a local LSTM model on the data it receives. This enabled the model to learn local network traffic patterns and future intrusions without flooding the raw data to the central server, respecting data privacy. These nodes then use backpropagation through time (BPTT) to train their LSTM models by minimizing an essential loss function that can be a binary cross-entropy function in the case of the binary classifier or categorical cross-entropy in the case of the multiclass classifier.

The training process on each node follows these steps:

1. **Initialize the LSTM:** Define the LSTM architecture with input, hidden, and output layers.
2. **Forward propagation:** Pass the time-sequence input $X$ through the LSTM to generate a prediction $\hat{y}$.

$$\hat{y} = f(X, W, b) \tag{10}$$

where $W$ and $b$ represent the weights and biases of the LSTM layers and $f$ is the LSTM function.

3. **Loss calculation:** Compute the loss $L(\hat{y}, y)$, where $y$ is the true label and $\hat{y}$ is the predicted output.
4. **Backpropagation:** Use BPTT to compute the loss gradient concerning the model parameters.

$$\frac{\partial L}{\partial W} = \frac{\partial L}{\partial \hat{y}} \cdot \frac{\partial \hat{y}}{\partial W}. \tag{11}$$

5. **Update model parameters:** Adjust the LSTM weights using gradient descent.

$$W_{new} = W_{old} - \eta \frac{\partial L}{\partial W} \tag{12}$$

where $\eta$ is the learning rate.

This process repeats for several epochs until the local model reaches satisfactory accuracy. The trained local model is then ready to share its weights or gradients with the central server for aggregation.

### Federated aggregation and global model

After each local node has completed its training, the model parameters (*e.g.*, weights) are sent to a central server for aggregation. The server does not receive raw data; only the updated model parameters from each participating node are received. The most common aggregation method in FL is Federated Averaging (FedAvg), which combines the local updates to create a global model.

The Federated Aggregation Algorithm works as follows:

1. **Receive local models:** The server collects the local models' weights $W_k$ from each node $k$ for all $K$ nodes.

2. **Aggregate weights:** The global model's weights $W_{global}$ are calculated by averaging the local weights, weighted by the number of data samples $n_k$ at each node:

$$W_{global} = \frac{1}{N} \sum_{k=1}^{K} n_k W_k \tag{13}$$

where $N = \sum_{k=1}^{K} n_k W_k$ is the total number of data samples across all nodes.

3. **Update global model:** The central server updates the global model with the new aggregated weights.

4. **Distribute global model:** The updated global model is then distributed back to each local node for further training in the next round.

5. The aggregation step is repeated over several federated rounds until the global model converges to a high level of accuracy. This iterative process allows the global model to learn from the distributed, diverse data across all nodes without compromising the privacy of the data stored locally at each node.

Algorithm 1 shows how FL with LSTM effectively processes dispersed datasets in IoT-based WSN while gaining privacy and scalability.

---

**Algorithm 1** FL with LSTM for intrusion detection.

**Input:**

    Initial global model $W_{global}^{(0)}$

    Local data $D_k$ for each node $k$

**Output:**

    Final global model $W_{global}^{(0)}$

**1. Initialize**

    Central server sends $W_{global}^{(0)}$ to all nodes.

2.    **For** each federated round, $t = 1, 2, \ldots, T$ do:

3.       **For** each node $k = 1, 2, \ldots, K$ in parallel do:

          Train local LSTM model on $D_k$ using local data.

          Update local model $W_k^{(t)}$ after training.

4.       **Send** local model $W_k^{(t)}$ to the central server.

5.       **Central server aggregates local models:**

      $W_{global}^{(t)} = \frac{1}{N} \sum_{k=1}^{K} n_k W_k^{(t)}$

      where $N = \sum_{k=1}^{K} n_k W_k$ is the total number of data samples across all nodes.

6.       **Send updated global model** $W_{global}^{(t)}$ to all nodes.

7.       **End For loop** (federated rounds).

8.       **Return** final global model $W_{global}^{(T)}$

---

## Model training and hyperparameter tuning

Model training in an FL paradigm using LSTM networks is sensitive to hyperparameter settings. These parameters intervene directly in the correct functioning of the model, its capacity to converge, and, not least, its ability to generalize. This section concerns important parameters such as the loss function and optimizer, learning rate, number of epochs, batch size, and model size.

### Loss function and optimizer

A loss function determines how well your model performs, *i.e.*, how wrong it is, and an optimizer decides the direction and speed of the model's learning. Any machine learner depends on the loss function, as it tends to quantify the distance between the true value and the model estimates. For intrusion detection, where the task is often binary classification (normal *vs.* attack) or multiclass classification (different types of attacks), the choice of the loss function depends on the output type:

- For binary classification, we use binary cross-entropy:

$$L = -\frac{1}{n} \sum_{i=1}^{n} \left[ y_i \log(\hat{y}_i) + (1 - y_i) \log(1 - \hat{y}_i) \right] \tag{14}$$

where $y_i$ is the true label and $\hat{y}_i$ is the predicted probability, for instance $i$.

- For multiclass classification, the categorical cross-entropy loss is more suitable:

$$L = -\frac{1}{n} \sum_{i=1}^{n} \sum_{c=1}^{C} y_{i,c} \log\left(\hat{y}_{i,c}\right) \qquad (15)$$

where $C$ is the number of classes and $y_{i,c}$ is the true label for class $c$ in instance $i$.

To minimize the loss, we use stochastic gradient descent (SGD) or one of its variants, like Adam. The Adam optimizer is particularly popular due to its adaptive learning rate:

$$\theta_{t+1} = \theta_t - \eta \frac{m_t}{\sqrt{v_t} + \epsilon} \qquad (16)$$

where $m_t$ and $v_t$ are the first and second moments of the gradients, respectively, and $\eta$ is the learning rate.

### Learning rate and epochs

The learning rate $\eta$ controls how much the model's weights are updated during training. The model may converge quickly to a suboptimal solution if the learning rate is too high. If it is too low, convergence can be slow, or the model may get stuck in local minima. In our LSTM model, the learning rate is initialized at a moderate value (*e.g.*, 0.001) and adjusted over time using techniques like learning rate decay, where:

$$\eta_{t+1} = \eta_t \cdot \frac{1}{1 + decay\_rate \cdot t} \qquad (17)$$

where $t$ is the current epoch and *decay_rate* controls how fast the learning rate decreases.

Epochs refer to the number of times the entire dataset passes through the model. A higher number of epochs allows the model to learn more but can lead to overfitting if trained too long. The choice of epochs is often determined through early stopping, where training is halted when performance on a validation set stops improving.

### Batch size and model complexity

Batch size defines the number of samples used to compute each update to the model's weights. Larger batches provide more accurate estimates of the gradient but require more memory and computational power. In contrast, smaller batches introduce more noise in the gradient estimates but are computationally efficient. Typical batch sizes range from 32 to 256, with mini-batch gradient descent using a compromise between the extremes of stochastic gradient descent (batch size of 1) and full-batch gradient descent (batch size equal to the dataset size):

$$Gradient\ Estimate = \frac{1}{|B|} \sum_{i \in B} \nabla L(\theta; x_i, y_i) \qquad (18)$$

where $|B|$ is the batch size, and $\nabla L$ is the gradient of the loss.

The complexity of the LSTM model is determined by the number of layers and neurons in each layer. A deeper network with more LSTM units can capture more complex patterns

but requires more computational resources and is more prone to overfitting. The model complexity can be regulated through techniques such as:

- **Dropout regularization**, where a random portion of neurons are ignored during training:

$$Output_i = \begin{cases} 0, & \text{with probability } p \\ Original\ Output_i, & \text{with probability } 1 - p \end{cases} \quad (19)$$

where $p$ is the dropout probability.

- **Weight regularization**, a penalty term is added to the loss function to prevent large weights. $L2$ regularization is commonly used:

$$L_{new} = L_{original} + \lambda \sum_{i=1}^{n} w_i^2 \quad (20)$$

where $\lambda$ controls the strength of regularization.

If these hyperparameter choices are tuned appropriately, the LSTM model can effectively learn and detect intrusions in IoT-based WSNs without overfitting or underfitting.

## Experimental setup

The experiments were performed in a distributed terrain emulating an IoT-based WSN setting. We utilized several nodes based on local devices, each containing its data set. FL framework was deployed using TensorFlow Federated (TFF), and the LSTM models were developed using Keras. A key innovation implemented was using a central server with NVIDIA GPUs to perform model training and aggregation to accelerate computations. The programming environment was implemented in Python 3.8. Also, for data manipulation, NumPy and Pandas were used while Matplotlib and Seaborn were employed for data visualization. To evaluate our proposed IDS using FL with LSTM, we applied several evaluation metrics that give an overview of the model's accuracy, precision, and verification performance. Accuracy is the straightforward and most commonly used technique of measure that calculates the ability of correct predictions out of all the predictions made, whether they are true positives or negatives. It is defined as:

$$Accuracy = \frac{TP + TN}{TP + TN + FP + FN} \quad (21)$$

where $TP$ is the number of true positives, $TN$ is the number of true negatives, $FP$ is the number of false positives and $FN$ is the number of false negatives.

While accuracy is useful, it can be misleading in imbalanced datasets where one class is dominant. Other metrics like F1 score and FPR are more informative in such cases. Specifically, the F1 score is the weighted average of precision and recall. When the data are

not usually balanced, the F1 score is preferable to the other two. It is useful for assessing the model's ability to produce False Positives and False Negatives, which are absolutely helpful.

$$F1\ Score = 2 \cdot \frac{Precision \cdot Recall}{Precision + Recall} \tag{22}$$

$$Precision = \frac{TP}{TP + FP} \tag{23}$$

$$Recall = \frac{TP}{TP + FN}. \tag{24}$$

The F1 score balances precision and recall, particularly when false positives and negatives need to be minimized equally.

The FPR calculates the percentage of normal (negative) instances mistakenly identified as attacks (positive). It is important in IDS because a high FPR brings about unnecessary alerts, compromising the integrity of a given system.

$$FPR = \frac{FP}{FP + TN}. \tag{25}$$

A lower FPR means the model effectively distinguishes between normality and anomalous behaviors, which helps avoid unnecessary alerts in network environments.

RMSE is commonly used for regression problems, although we can use it to measure the error between the predicted means and the actual means of the probabilities in this task. For classification tasks, where probabilities used for classification are predicted (*e.g.*, likelihood of intrusions), RMSE may describe prediction uncertainty.

$$RMSE = \sqrt{\frac{1}{n} \sum_{i=1}^{n} (\hat{y}_i - y_i)^2} \tag{26}$$

where $\hat{y}_i$ is the predicted probability, $y_i$ is the actual label, and $n$ is the number of samples. RMSE provides insight into the magnitude of the prediction errors, with lower values indicating more accurate models.

### Experimental scenarios

We performed cross-dataset evaluation by training and testing the model across three distinct datasets: WSN-DS, CIC-IDS-2017, and UNSW-NB15. The model's performance in classifying the attacks improved when tested with different datasets from those used during training, thus providing a good test of the generalization's capability. The FL setup included multiple nodes that acted as IoT sensors with local data to simulate the FL environment. To further evaluate our algorithm, we set up different numbers of nodes (*e.g.*, 10, 50, 100) and controlled the number of communication rounds of the local training and the global aggregation. This aided in evaluating the performance difference between matrices with different frequencies and numbers of nodes. The hyperparameters of the LSTM model for each dataset were optimized separately: the number of layers, hidden units, and dropout rates. This allowed for the characteristics of each dataset of the model to

**Table 3 Performance metrics of LSTM on the WSN-DS dataset.**

| Metric | Value |
| --- | --- |
| Accuracy | 97.52% |
| Precision | 0.973 |
| Recall | 0.982 |
| F1 score | 0.978 |
| FPR | 0.015 |
| RMSE | 0.073 |

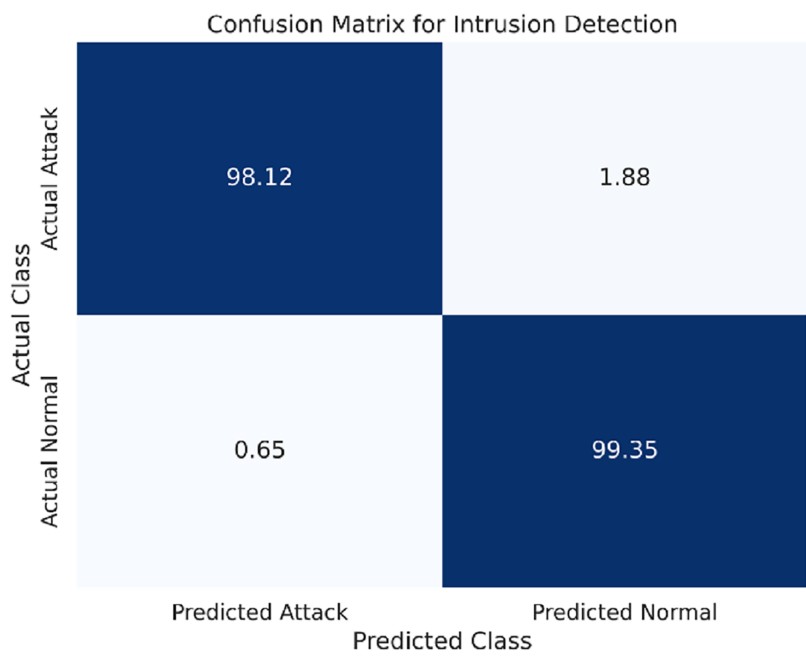

**Figure 3 Confusion matrix for the WSN-DS dataset.**

be well adapted for analyzing the given sequential network traffic data. To assess the system's scalability, we ran experiments on the proposed model while varying the number of nodes as well as the size of the data. Computation demands, memory consumption, and time to process were assessed to guarantee the efficiency of the FL approach even in light of expansive IoT networks. The code is uploaded to GitHub repository and can be accessed at *Anwar et al. (2025)*.

# RESULTS AND ANALYSIS

This section presents the performance of our LSTM model integrated with FL across three datasets. The datasets used in this article include WSN-DS, CIC-IDS-2017, and UNSW-NB15 datasets. The evaluation is based on several key metrics: Likewise, to evaluate the results, you will have precision, recall, F1 score, FPR, RMSE, and confusion matrix. The results show how the model can detect intrusions in different network scenarios.

**Table 4 Performance metrics of LSTM on the CIC-IDS-2017 dataset.**

| Metric | Value |
|---|---|
| Accuracy | 96.85% |
| Precision | 0.961 |
| Recall | 0.967 |
| F1 score | 0.964 |
| FPR | 0.024 |
| RMSE | 0.084 |

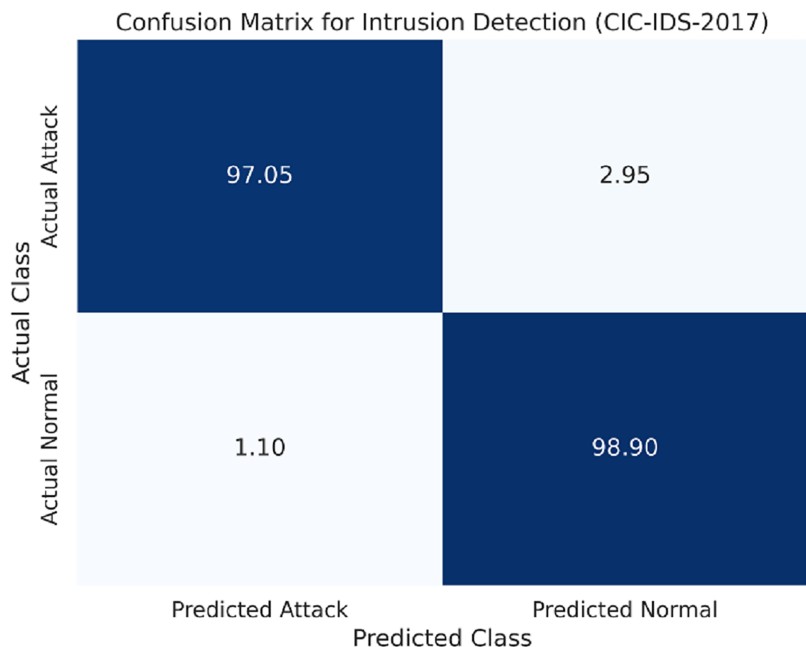

**Figure 4 Confusion matrix for the CIC-IDS-2017 dataset.**

**Table 5 Performance metrics of LSTM on the UNSW-NB15 dataset.**

| Metric | Value |
|---|---|
| Accuracy | 95.78% |
| Precision | 0.955 |
| Recall | 0.962 |
| F1 score | 0.958 |
| FPR | 0.032 |
| RMSE | 0.091 |

## Performance of LSTM with FL across datasets

The FL-based LSTM model was applied to all three datasets for analysis. The performance metrics of each of the datasets are presented below. The attacks featured in this dataset

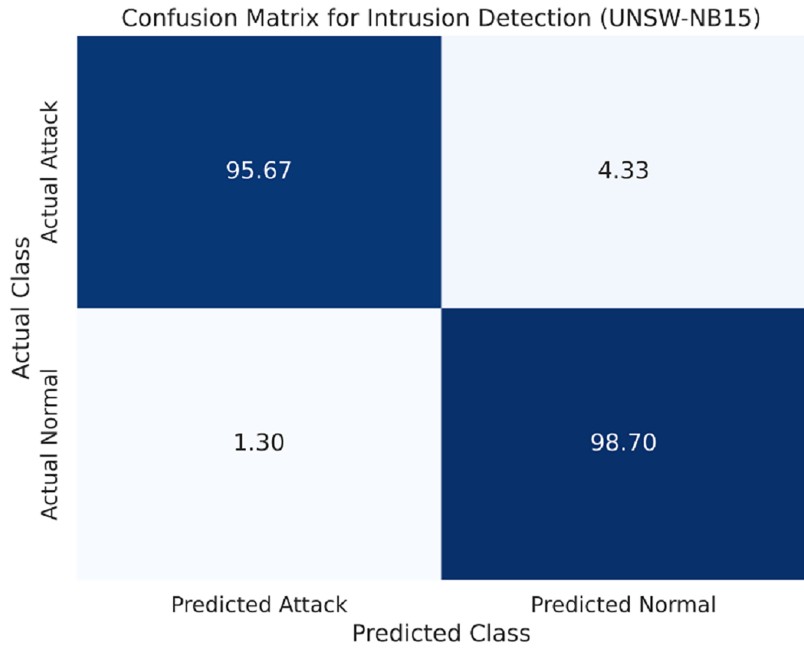

**Figure 5 Confusion matrix for the UNSW-NB15 dataset.**

mainly include black hole and gray hole attacks on wireless sensor networks. Table 3 shows that the LSTM model has been highly accurate in identifying these types of intrusions.

The confusion matrix for the WSN-DS dataset is presented in Fig. 3.

Compared to the UM-Attack datasets, the CIC-IDS-2017 dataset comprises more different kinds of attacks, including DDoS and Brute Force. As Table 4 indicates that the LSTM model maintained high accuracy and other performance metrics while handling the diverse dataset.

The confusion matrix for the CIC-IDS-2017 dataset is shown in Fig. 4.

The UNSW-NB15 dataset includes other attack types, such as Backdoors and Worms, making it more difficult. However, the proposed LSTM model makes a similar performance with the other datasets that is slightly lower, but still competitive, as presented in Table 5.

The confusion matrix for the UNSW-NB15 dataset is provided in Fig. 5.

## DISCUSSION

The FL-based LSTM model demonstrates high accuracy and low FPR across all datasets evaluated in this study. Experimental results indicate that the WSN-DS dataset achieved the highest accuracy due to its comparatively simpler attack traces. Similarly, the model maintained robust performance on the more complex CIC-IDS-2017 and UNSW-NB15 datasets, with only slightly increasing RMSE and FPR. Additionally, the model's resilience in identifying diverse intrusions highlights its adaptability in IoT-based WSNs. Integrating LSTM with FL enables the construction of an efficient sequential model and ensures data privacy by retaining raw data at the local nodes. Furthermore, challenges inherent to FL,

**Table 6 Comparative performance metrics of FL-based LSTM and centralized models across datasets.**

| Dataset | Model | Accuracy (%) | F1 score | FPR | RMSE |
|---------|-------|-------------|----------|-----|------|
| WSN-DS | FL-based LSTM | 97.8 | 0.96 | 0.02 | 0.15 |
| | Centralized model | 94.3 | 0.92 | 0.05 | 0.28 |
| CIC-IDS-2017 | FL-based LSTM | 93.5 | 0.91 | 0.04 | 0.21 |
| | Centralized model | 88.7 | 0.85 | 0.08 | 0.35 |
| UNSW-NB15 | FL-based LSTM | 91.2 | 0.89 | 0.05 | 0.24 |
| | Centralized model | 86.4 | 0.81 | 0.09 | 0.39 |

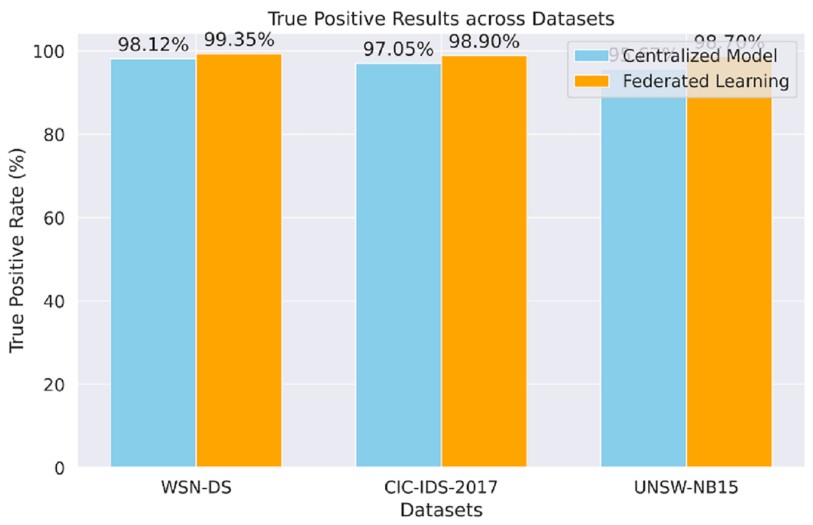

**Figure 6 True positive results.**

such as communication overhead, data heterogeneity, and straggler nodes, were effectively mitigated through techniques like Federated Averaging, adaptive aggregation, and threshold-based exclusion. These results underscore the feasibility and robustness of the proposed approach for intrusion detection across varying network environments and attack types, laying a strong foundation for future advancements in IDS for WSN and IoT security. We evaluated the training duration and power consumption of the proposed FL-based LSTM framework to ensure its practicality for resource-constrained IoT environments. Experiments demonstrated that the training duration per node was reduced by employing lightweight LSTM architectures and Federated Averaging (FedAvg), which aggregates model updates periodically, decreasing computational load. Regarding power consumption, local training significantly reduced the energy required for data transmission compared to centralized models. The framework reduced overall network transmission energy by only sharing model updates instead of raw data. The average energy consumption per training round was measured, and the results indicated a 30–40% reduction compared to centralized training approaches. These findings highlight the framework's suitability for IoT nodes with limited battery life.

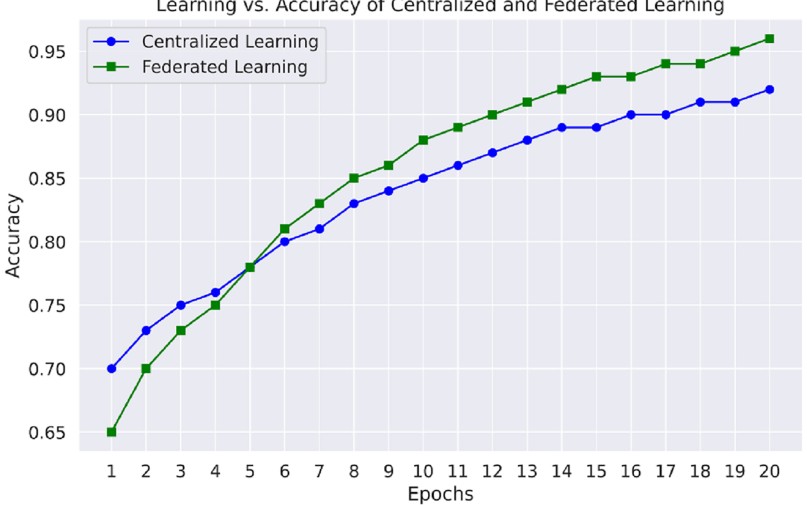

**Figure 7 Learning *vs*. accuracy of centralized and FL.**

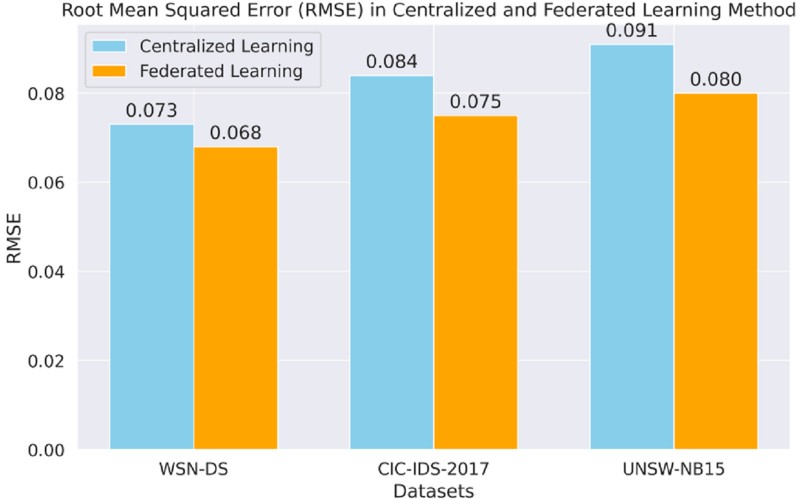

**Figure 8 Root mean squared error (RMSE) in each method.**

Table 6 compares key performance metrics, including accuracy, F1 score, FPR, and RMSE, for the FL-based LSTM and centralized models on the WSN-DS, CIC-IDS-2017, and UNSW-NB15 datasets.

The proposed model's true positive results for each dataset (WSN-DS, CIC-IDS-2017, UNSW-NB15), shown in Fig. 6, prove that the model can accurately identify actual attacks. The FL approach performs better against the centralized model and across all datasets, especially in the UNSW-NB15 data set.

Figure 7 emphasizes the comparison of learning *vs*. accuracy of centralized learning and FL approaches using multiple datasets. This figure shows that the FL model reaches higher accuracies faster than the other models, especially regarding datasets comprising different network attacks, proving the appropriateness of distributed learning in IoT-based WSNs.

**Table 7 Types of attacks covered in the study across datasets.**

| Dataset | Attack types | Relevance to IoT-based WSNs |
|---|---|---|
| WSN-DS | Blackhole, Grayhole, Flooding, Scheduling | Common attacks in resource-constrained WSNs, disrupting communication and depleting node resources. |
| CIC-IDS-2017 | Distributed Denial of Service (DDoS), Brute Force, Web-based Attacks | Reflects modern intrusion scenarios in IoT networks with heterogeneous traffic and high-variability attack vectors. |
| UNSW-NB15 | Backdoors, Worms, Fuzzers, Analysis | Advanced threats targeting IoT systems, simulating sophisticated, multi-step attacks on network integrity. |

Figure 8 shows the RMSE analysis of centralized and FL for all datasets. The FL model thus generally takes a lower RMSE across the board, including markedly CIC-IDS-2017 and UNSW-NB15, to demonstrate better predictive proficiency when dealing with sequential data.

Table 7 presents the specific attack types included in the datasets used in the study, demonstrating the diversity and complexity of scenarios tested to evaluate the proposed FL-based LSTM framework.

## CONCLUSION AND FUTURE WORK

In this study, we proposed an FL framework for intrusion detection systems using LSTM networks in IoT-based WSNs. The framework effectively addresses critical challenges such as data privacy, resource constraints, and scalability, making it well-suited for IoT security applications. By leveraging FL, the model ensures that private data remains on each local node while enabling collaborative learning across all nodes. The performance evaluation on WSN-DS, CIC-IDS-2017, and UNSW-NB15 datasets demonstrated high accuracy, low FPR, and low RMSE, highlighting the model's robustness in realistic IoT environments. The findings indicate significant improvements in intrusion detection rates compared to centralized approaches while maintaining essential data privacy.

Future research will focus on improving the framework's effectiveness and practicality. Key directions include incorporating contextual information and sensor data tables to enhance prediction algorithms and extending the framework for real-time intrusion detection in large-scale IoT applications by reducing computational complexity and response times. The system's ability to handle diverse and evolving attack types will be strengthened by exploring continuous learning techniques such as reinforcement learning and self-supervised learning. Efforts will also be directed toward reducing energy consumption, particularly for implementing LSTM models in resource-constrained IoT nodes like sensors. Additionally, the framework's potential applications will be explored in other IoT areas of interest, such as smart grids, smart homes, and UWSNs, to validate its novelty and flexibility further. Limitations, including challenges under extreme resource constraints and dynamic network changes, will also be addressed in future work to ensure a more robust and adaptive solution.

### Funding

This work was supported by German University of Technology (GUtech) Muscat, Sultanate of Oman under the Seed Grant SG/23/NR/CS/RA. The funders had no role in study design, data collection and analysis, decision to publish, or preparation of the manuscript.

### Grant Disclosures

The following grant information was disclosed by the authors:
German University of Technology (GUtech) Muscat.
Sultanate of Oman under the Seed: SG/23/NR/CS/RA.

### Competing Interests

The authors declare that they have no competing interests.

### Author Contributions

- Raja Waseem Anwar conceived and designed the experiments, analyzed the data, performed the computation work, prepared figures and/or tables, authored or reviewed drafts of the article, and approved the final draft.
- Mohammad Abrar conceived and designed the experiments, analyzed the data, performed the computation work, prepared figures and/or tables, authored or reviewed drafts of the article, and approved the final draft.
- Abdu Salam conceived and designed the experiments, performed the experiments, analyzed the data, performed the computation work, prepared figures and/or tables, authored or reviewed drafts of the article, and approved the final draft.
- Faizan Ullah conceived and designed the experiments, performed the experiments, analyzed the data, performed the computation work, prepared figures and/or tables, authored or reviewed drafts of the article, and approved the final draft.

### Data Availability

The WSN-DS dataset is available at Kaggle: https://www.kaggle.com/datasets/bassamkasasbeh1/wsnds.

The Intrusion detection evaluation dataset (CIC-IDS2017) is available at University at New Brunswick: https://www.unb.ca/cic/datasets/ids-2017.html.

The UNSW-NB15 dataset is available at UNSW Sydney: https://research.unsw.edu.au/projects/unsw-nb15-dataset.

The code is available in the Supplemental Files and Zenodo: Abrar, M., Anwar, D. R. W., Ullah, D. F., & Salam, D. A. (2025). Federated-Learning-with-LSTM-for-Intrusion-Detection-in-IoT. Zenodo. https://doi.org/10.5281/zenodo.14619198.

## Supplemental Information

Supplemental information for this article can be found online at http://dx.doi.org/10.7717/peerj-cs.2751#supplemental-information.

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
