# Peer review of "Federated learning with LSTM for intrusion detection in IoT-based wireless sensor networks: a multi-dataset analysis"

_PeerJ Computer Science, doi:10.7717/peerj-cs.2751_

## Round 0.1 · original submission · Major Revisions

I have received reviews of your manuscript from scholars who are experts on the cited topic. They find the topic interesting; however, several concerns must be addressed regarding experimental setup, performance metrics, datasets, comparisons, computational complexity, and presentation. These issues require a major revision. Please refer to the reviewers’ comments at the end of this letter; you will see that they advise you to revise your manuscript. If you are prepared to undertake the work required, I would be pleased to reconsider my decision. Please submit a list of changes or a rebuttal against each concern when you submit your revised manuscript.

Thank you for considering PeerJ Computer Science for the publication of your research.

With kind regards,

Reviewer 1 ·

Basic reporting

no comment

Experimental design

no comment

Validity of the findings

no comment

Additional comments

The suggestions for improving the manuscript are given as follows:
1. While the paper integrates FL with LSTM, it would be beneficial to justify why LSTM is particularly suited for intrusion detection in IoT-based WSNs. Are the sequential patterns in intrusion data a primary factor? Including a comparison with other deep learning models, like GRU or CNN, could strengthen the argument for choosing LSTM.
2. The text mentions FL for privacy and scalability but does not address potential challenges such as communication overhead, data heterogeneity, or straggler nodes in IoT environments. Discussing these limitations and how they were mitigated would add depth to the study.
3. The inclusion of three datasets (WSN-DS, CIC-IDS-2017, UNSW-NB15) is commendable. However, it would be helpful to explain the specific relevance of these datasets to IoT-based WSNs. Are there unique IoT-specific challenges (e.g., low bandwidth, resource constraints) captured by these datasets?
4. While the paper evaluates metrics like accuracy, F1 score, FPR, and RMSE, it might be useful to include additional metrics such as precision, recall, or AUC-ROC to provide a more comprehensive performance assessment, especially for imbalanced datasets where FPR might not tell the full story.
5. The text notes improvements over centralized models but does not quantify them or provide detailed comparative results. Including a table or graph comparing the performance metrics (e.g., accuracy, FPR) of the FL-based LSTM model and centralized models would clarify the extent of the improvement.
6. IoT devices are typically resource-constrained. How does the proposed FL-based LSTM framework address limitations in memory, computation, and battery life at the IoT nodes? Was resource consumption evaluated during the experiments?
7. The text mentions "complex attack scenarios" but does not provide examples. Highlighting specific types of attacks (e.g., DoS, DDoS, spoofing) tested in the study would demonstrate the framework's robustness.
8. The study claims scalability, but there is no discussion about the size of the IoT network used for evaluation. Was the framework tested in large-scale scenarios with many IoT nodes? Detailing the experimental setup and the number of participating nodes would substantiate this claim.
9. The study lacks a detailed discussion of limitations and areas for future work. For instance, how might the framework perform under extreme resource constraints or dynamic network changes? Future work could explore combining FL with lightweight models or enhancing robustness against adversarial attacks.
10. Federated Learning requires periodic model updates between nodes and the central server. Did the study evaluate communication costs, particularly in bandwidth-constrained IoT environments?
11. While the framework shows promise, its real-world applicability would benefit from discussion. How does it perform in live IoT networks with noisy or missing data? Was the model tested under real-time constraints?

·

Basic reporting

The writing in this paper is generally professional and easy to follow. The authors have done a good job providing adequate background and context, especially in the field of IoT wireless sensor networks and intrusion detection systems. The literature references are sufficient, and the structure of the article is professional. Datasets and preprocessing steps are introduced thoroughly, ensuring the reproducibility of the experiments. All terms and concepts are clearly defined, contributing to the overall clarity of the paper.

Experimental design

The paper proposes a novel intrusion detection system using a federated learning framework with LSTM networks, which fits well within the journal's aims. The research question is relevant and meaningful, addressing a knowledge gap in IoT security. The authors have described the methods in detail, making replication possible.

Validity of the findings

While the detection results are impressive in absolute terms, the impact and novelty of the findings could be better assessed by including baseline models for comparison. This would provide a clearer understanding of the proposed model's effectiveness.

Additionally, the practicality of training LSTM models on resource-limited IoT sensors raises questions about training duration and power consumption. The conclusions drawn are well stated and linked to the original research question, but further exploration of the proposed model's resource efficiency and network transmission benefits compared to centralized frameworks would strengthen the findings.

Also, it would be more convincing to align the experimental setups with real-world scenarios, such as those mentioned in the introduction like smart cities or smart healthcare, by considering the number and types of sensors used in those settings.

Additional comments

Overall, the paper is well-written and presents a novel approach to intrusion detection in IoT wireless sensor networks. The proposed system is well-explained, but addressing practical concerns like node heterogeneity and the capability of resource-limited IoT sensors to train the LSTM model would enhance its practicality. Experiments showcasing the benefits of federated learning in terms of resource usage and network transmission would be beneficial.

---

## Round 0.2 · accepted · Accept

I am pleased to inform you that your work has now been accepted for publication in PeerJ Computer Science.

Please be advised that you cannot add or remove authors or references post-acceptance, regardless of the reviewers' request(s).

Thank you for submitting your work to this journal. I look forward to your continued contributions on behalf of the Editors of PeerJ Computer Science.

With kind regards,

·

Basic reporting

The writing is clear and professional, effectively addressing all previous concerns. The background and context in IoT wireless sensor networks and intrusion detection systems are well-established. The article is well-structured, with sufficient literature references, and the datasets and preprocessing steps ensure reproducibility.

Experimental design

This paper presents original research that fits within the journal's Aims and Scope. The research question is clearly defined and relevant, addressing a significant knowledge gap in IoT security. The methods are described in sufficient detail for replication.

Validity of the findings

The impact and novelty of the findings have been adequately assessed in this revision. The authors have included baseline models for comparison and showed the effectiveness of the proposed model. The conclusions drawn are well stated and directly linked to the original research question, remaining within the bounds of the supporting results. Overall, the findings are robust and statistically sound.

Additional comments

Overall, the paper is well-written and presents a novel approach to intrusion detection. The authors have successfully addressed all previous comments, and I have no further concerns.